# Epigallocatechin-3-Gallate Enhances Antioxidant Activity and Improves Testicular and Epididymal Histology in Cadmium-Exposed Prepubertal Rats

**DOI:** 10.3390/ijms26178264

**Published:** 2025-08-26

**Authors:** Sonia Guadalupe Pérez-Aguirre, Rosa María Vigueras-Villaseñor, Herlinda Bonilla-Jaime, Sergio Montes, Sonia Galván-Arzate, Joel Hernández-Rodríguez, Sergio Marín de Jesús, Leticia Carrizales-Yañez, Julio Cesar Rojas-Castañeda, Marcela Arteaga-Silva

**Affiliations:** 1Doctorado en Ciencias Biológicas y de la Salud, Universidad Autónoma Metropolitana, Ciudad de México C.P. 09340, Mexico; soniagaguirre@gmail.com (S.G.P.-A.);; 2Laboratorio de Biología de la Reproducción, Instituto Nacional de Pediatría, Av. Insurgentes Sur 3700-Letra C, Insurgentes Cuicuilco, Coyoacán, Ciudad de México C.P. 04530, Mexico; 3Departamento de Biología de la Reproducción, División de Ciencias Biológicas y de la Salud, Universidad Autónoma Metropolitana-Iztapalapa, Av. Ferrocarril San Rafael Atlixco 186, Col. Leyes de Reforma 1ª. Sección, Alcaldía Iztapalapa, Ciudad de México C.P. 09340, Mexico; 4Laboratorio de Química Analítica, Unidad Académica Multidisciplinaria Reynosa-Aztlan, Universidad Autónoma de Tamaulipas, Calle 16 y Lago de Chapala, Col. Aztlan Reynosa, Reynosa Tamaulipas C.P. 88740, Mexico; 5Departamento de Neuroquímica, Instituto Nacional de Neurología y Neurocirugía, Manuel Velasco Suárez, Insurgentes Sur 3877, Col. La Fama, Ciudad de México C.P. 14269, Mexico; 6Cuerpo Académico en Quiropráctica, Universidad Estatal del Valle de Ecatepec, EDOMEX, A. Central s/n esq. Leona Vicario Col. Valle de Anáhuac Secc. A Ecatepec de Morelos, Estado de México C.P. 55210, Mexico; 7Coordinación de Innovación y Aplicación de la Ciencia y la Tecnología CIACYT, Facultad de Medicina, Universidad Autónoma de San Luis Potosí, San Luis Potosí C.P. 78210, Mexico; 8Laboratorio de Neuroendocrinología Reproductiva, Departamento de Biología de la Reproducción, Universidad Autónoma Metropolitana-Iztapalapa, Av. Ferrocarril San Rafael Atlixco 186, Col. Leyes de Reforma 1ª. Sección, Alcaldía Iztapalapa, Ciudad de México C.P. 09340, Mexico

**Keywords:** epigallocatechin-3-gallate, cadmium, puberty, antioxidant enzymes, testes, epididymides

## Abstract

The prepubertal period represents a critical stage of development, where the reproductive system is susceptible to toxicants such as cadmium (Cd). Cd induces oxidative stress, causes alterations in the antioxidant enzymes and testosterone concentration, and affects reproductive organs. (–)-Epigallocatechin-3-gallate (EGCG), a polyphenol with antioxidant properties, has been studied for its protective effects. We evaluated the effects of EGCG on antioxidant activity and improvement of testicular and epididymal histology in Cd-exposed prepubertal rats. Twenty-eight male Wistar rats, on postnatal day (PND) 21, were distributed into four groups: Ctrl (saline), Cd (1 mg/kg CdCl_2_), EGCG (10 mg/kg), and Cd+EGCG (1 mg/kg CdCl_2_ + 10 mg/kg EGCG). Treatments were administered intraperitoneally from PND 21 to 49. After euthanasia, blood, testes, and epididymides were collected for Cd content, testosterone concentration, antioxidant activity, and histological evaluation. Cd exposure increased blood Cd, reduced testosterone, impaired antioxidant activity, and caused epithelial disorganization in both organs. In contrast, co-administration of EGCG significantly lowered Cd accumulation, restored testosterone concentration and antioxidant enzymes, and preserved histological integrity of testes and epididymides. These findings demonstrate that EGCG exerts protective effects against Cd-induced reproductive damage during the prepubertal period, suggesting its potential therapeutic use to counteract Cd toxicity in reproductive development.

## 1. Introduction

The prepubertal stage is a period that precedes puberty. This stage is a critical period for the development of the testes and epididymides, as well as other reproductive organs. This period prepares the individual to begin puberty. Puberty is a transitional phase between childhood and adolescence, characterized by endocrine and morphological changes that are essential for reaching sexual maturity in the adult stage [1,2]. The onset of puberty is marked by an increase in the secretion of kisspeptin and neurokinin B, which leads to a resurgence of pulsatile gonadotropin-releasing hormone (GnRH) signaling from the hypothalamus. This stimulates increased pituitary secretion of luteinizing hormone (LH) and follicle-stimulating hormone (FSH), which stimulate the production of gonadal sex hormones. This promotes the timely activation and precise regulation of the hypothalamic–pituitary–gonadal (HPG) axis [3]. This process occurs in various species, including humans, non-human primates, and rodents [4,5]. In male rats, puberty occurs around postnatal day (PND) 33–55, after the prepubertal stage from PND 21 to 32 [6]. The development and function of the testes and epididymides are important during puberty, when testosterone concentrations rise, initiating the first spermatogenic wave [7,8], while the epididymis completes its cell differentiation and expansion [7,8]. Although the timing of puberty is largely programmed by genetics, the entire process can be disrupted from the prepubertal stage by environmental toxins, such as cadmium (Cd). This heavy metal primarily results from human activities, with mining and the metallurgical, textile, and automotive industries being the main sources [9]. In addition, the use of pesticides, insecticides, and fertilizers contaminates food and drinking water. According to the Occupational Safety and Health Administration (OSHA) of the United States of America, the permissible exposure limit (PEL) for occupational exposure to Cd or Cd oxide is 0.1 mg/m^3^ in a workday [10]. Meanwhile, the World Health Organization (WHO) [11] establishes that the tolerable monthly ingestion of Cd is 25 µg/kg. Cd interferes with cellular functionality, proliferation, and differentiation mainly through the formation of reactive oxygen species (ROS), which lead to oxidative stress [12,13]. The resulting oxidative stress leads to damage to proteins, lipids, and DNA [14,15,16,17,18]. This is one of the mechanisms by which Cd can impair the activity of antioxidant enzymes, such as superoxide dismutase (SOD) and its isoforms (Cu/Zn-SOD and Mn-SOD), catalase (CAT), and glutathione peroxidase (GPx), in the testes [19,20,21] and epididymides [21,22] of pubertal and adult rats, suggesting that the antioxidant defense is specific to these organs and is the result of their adaptive response to oxidative stress to maintain the function of these reproductive organs [19,23]. Cd causes damage to the structure of the testicular and epididymal epithelium, such as cellular disorganization and epithelial desquamation, loss of germ cells, tubular degeneration with vacuoles, apoptosis of spermatogenic cells [20,24], lack of adhesion between principal cells and basal cells, and a reduction in testosterone concentration [22]. To counteract these effects, various antioxidant molecules have been used, such as (-)-Epigallocatechin-3-gallate (EGCG), a polyphenolic compound and catechin found in the young leaves, shoots, and stems of the *Camellia sinensis* plant. It makes up 50 to 80% of the total catechin content of green tea and is the most active compound [25]. EGCG has been reported to reduce the toxicity of heavy metals like nickel (Ni), arsenic (As), lead (Pb), mercury (Hg), and Cd in tissues of organs such as the testicles, liver, kidneys, and neuronal cells [25,26]. One of the properties attributed to EGCG is its antioxidant capacity, which involves elimination of ROS and quelation of ionic metals [23,27,28]. Its high antioxidant activity depends on the presence of three hydroxyl groups (-OH) in its molecular structure [23]. The antioxidant mechanism involves the donation of hydrogen atoms from the -OH groups to neutralize free radicals [25]. Chelation occurs when these groups react with metal ions to form stable, ring-shaped structures called cyclic chelates. This structure increases the water solubility of EGCG, allowing it to be excreted in urine and eliminating heavy metals from tissues [23,25,26]. Thus, the Cd that impacts reproductive processes could be mitigated by EGCG, which could favor reproductive health. In the present study, the effect of EGCG on antioxidant activity and improvement of testicular and epididymal histology in Cd-exposed prepubertal rats was evaluated.

## 2. Results

### 2.1. Effects of Cd, EGCG, and Cd+EGCG on the Weight and Concentration of Cd in Blood, Testes, and Epididymides

Table 1 shows the weight of the reproductive organs (g). A significant decrease was found among the Cd group compared to the Ctrl, EGCG, and Cd+EGCG groups (*p* < 0.05). In blood, testes, and both regions of the epididymides, a higher concentration was observed in Cd compared to the Ctrl group (*p* < 0.05). In contrast, a significant decrease in Cd concentrations was observed in the blood, testes, and epididymides of subjects from the EGCG and Cd+EGCG groups compared to the Cd group (*p* < 0.05).

### 2.2. Effects of Cd, EGCG, and Cd+EGCG on Testosterone Concentrations

As shown in Table 2, the serum testosterone concentration showed a significant decrease in the Cd group compared to the Ctrl group (*p* < 0.05). Maintenance of testosterone concentration was observed in the EGCG and Cd+EGCG groups compared to the Cd group.

### 2.3. Effects of Cd, EGCG, and Cd+EGCG on Lipoperoxidation

Figure 1 shows that Malondealdehyde (MDA) concentration, a marker of lipid peroxidation, was significantly increased in the testes (*p* < 0.05), caput (*p* < 0.05), and cauda (*p* < 0.05) of the epididymides regions in the Cd group compared to the Ctrl group. The EGCG showed a significant decrease in MDA concentration in the testes (*p* < 0.05), caput (*p* < 0.05), and cauda (*p* < 0.05) of the epididymides regions compared to the Cd group. The Cd+EGCG showed a significant decrease in MDA concentration in the testes (*p* < 0.05), caput (*p* < 0.05), and cauda (*p* < 0.05) of the epididymides regions compared to the Cd group.

### 2.4. Effects of Cd, EGCG, and Cd+EGCG on Antioxidant Activity

Figure 2 shows the total SOD activity and its isoforms in the testes, caput, and cauda of the epididymides. In the testes (Figure 2A), a significant decrease was observed in the Cd group compared to the Ctrl group (*p* < 0.05). The EGCG and Cd+EGCG groups showed higher total SOD activity compared to the Cd group (*p* < 0.05). No differences were observed between the EGCG and Cd+EGCG groups when compared to the Ctrl group. Regarding the activity of the Cu/Zn-SOD isoform and the Mn-SOD isoform, a significant decrease in the activity of both was observed in the Cd group (*p* < 0.05) when compared to the Ctrl, EGCG, and Cd+EGCG groups. Regarding the total SOD activity in the caput epididymides region (Figure 2B), a significant decrease was observed in the Cd group when compared to the Ctrl, EGCG, and Cd+EGCG groups (*p* < 0.05). In the activity of the Cu/Zn-SOD and Mn-SOD isoforms, a decrease in the activity of both isoforms was observed in the Cd group when compared to the Ctrl, EGCG, and Cd+EGCG groups (*p* < 0.05). In the cauda region of epididymides (Figure 2C), a significant decrease in total SOD activity was observed in the Cd group when compared to the Ctrl, EGCG, and Cd+EGCG groups (*p* < 0.05). The activity of the Cu/Zn-SOD and Mn-SOD isoforms was observed to decrease in the Cd group when compared with the Ctrl, EGCG, and Cd+EGCG groups (*p* < 0.05). However, no alteration in the activity of either isoform was observed when the EGCG and Cd+EGCG groups were compared with the Ctrl group.

Figure 3 shows the effect of Cd, EGCG, and Cd+EGCG on CAT enzyme activity in the testes, caput, and cauda of the epididymides. A significant decrease in CAT activity was observed in the Cd group compared to the Ctrl group in the testes, caput, and cauda of the epididymides (*p* < 0.05). However, CAT enzyme activity in the EGCG and Cd+EGCG groups showed a significant increase compared to the Cd group in the caput and cauda of the epididymides (*p* < 0.05).

Figure 4 shows the effect of Cd, EGCG, and Cd+EGCG on GPx enzyme activity in the testes, caput, and cauda of the epididymides. A significant decrease in GPx activity was observed in the Cd group compared to the Ctrl group for the testes (*p* < 0.05), caput (*p* < 0.05), and cauda (*p* < 0.05) of the epididymides. In addition, the EGCG group showed a significant increase in GPx activity compared to the Cd group for the testes (*p* < 0.05), caput (*p* < 0.05), and cauda (*p* < 0.05) of the epididymides, while Cd+EGCG also showed a significant increase in GPx activity compared to the Cd group in the testes (*p* < 0.05), caput (*p* < 0.05), and cauda (*p* < 0.05) of the epididymides.

### 2.5. Effects of Cd, EGCG, and Cd+EGCG on Testicular Epithelium

Figure 5 shows the histological structure of the seminiferous tubules of the testicular epithelium under different treatments. The Ctrl group exhibited a stratified epithelium, with Sertoli cells supporting various stages of germ cells development, including spermatogonia, spermatocytes, and round and elongated spermatids, suggesting complete spermatogenesis. The basement membrane appeared intact and unaltered. In the lumen of the tubules, sperm were observed (Figure 5A). In the Cd epithelium of seminiferous tubules, desquamation of the cells, folding of the basal membrane, and hypertrophy of the spermatocytes were observed. In the lumen of the tubules, few sperm were observed (Figure 5B). In the EGCG seminiferous tubules, all types of germ cells at different stages of development were present, without basal membrane alterations, and a regular tubular structure was observed. In the lumen, a regular number of sperm were observed (Figure 5C). In Cd+EGCG, the seminiferous tubules exhibited all types of germ cells at different stages of development; however, an increase in the number of round spermatids was noted (Figure 5D).

In the interstitial zone, blood vessels, fibroblasts, and Leydig cells were present (Figure 5E). In the Cd group, the endothelial layers showed dilated blood vessels and atrophied Leydig cells (Figure 5F). The EGCG showed regular endothelial layers and Leydig cells without alterations (Figure 5G). In Cd+EGCG, a regular interstitial zone with blood vessels, fibroblasts, and Leydig cells (Figure 5H) was observed.

Table 3 shows the values of the maturation and the histopathological index, as well as the area and diameter of the seminiferous tubules. The diameter and area of the seminiferous tubules decreased significantly compared to the Ctrl, EGCG, and Cd+EGCG groups (*p* < 0.05). No alterations were found in the maturation or histopathological index across the different treatments.

### 2.6. Effects of Cd, EGCG, and Cd+EGCG on Epididymal Epithelium

Figure 6 shows the histological analysis of the caput and cauda regions of the epididymides. In the Ctrl group, the epididymides exhibited a pseudostratified epithelium composed of different cell types, including principal, clear, and basal cells, along with the presence of sperm in the lumen of the tubules in the caput (Figure 6A) and cauda (Figure 6E) of the epididymides. In the Cd group, there was cellular disorganization, a cribriform appearance, and increased vacuolization, along with dense vesicles in the cytoplasm. However, these changes did not affect the presence of sperm in the lumen (Figure 6B,F). In the EGCG group, principal, clear, and basal cells were noted, accompanied by moderate vacuolization. Additionally, sperm were observed in the lumen of the caput (Figure 6C) and cauda (Figure 6G) of the epididymides, like the Ctrl group. In the Cd+EGCG group, the caput (Figure 6D) showed the presence of different cell types, vacuolization, and high density, as well as sperm in the epididymal lumen. However, in the cauda (Figure 6H), slight hyperplasia was observed, which increased the height of the epithelium and revealed a slight increase in vacuolization compared to the Ctrl group. Nonetheless, these changes were more pronounced compared to the Cd group, and the presence of sperm in this region remained affected.

Table 4 displays the height and area of the principal cells of the epididymal epithelium. In the Cd group, a significant increase in the height and area of the principal cells in the caput and cauda of the epididymides was observed (*p* < 0.05) when compared to the Ctrl, EGCG, and Cd+EGCG groups. In the EGCG and Cd+EGCG groups, a significant decrease in the height and area of the principal cells was noted (*p* < 0.05) compared to the Cd group (similar values were present in the Ctrl group). Additionally, differences in the height and area of the principal cells were observed in both regions of the epididymides in the EGCG and Cd+EGCG groups compared to the Ctrl group.

## 3. Discussion

The EGCG is found in the young leaves, buds, and stems of the *Camellia sinensis* plant. It accounts for 50–80% of the total catechin content in green tea. EGCG is the most active compound [25] used to reduce the toxicity of heavy metals such as Ni, As, Pb, Hg, and Cd in organ tissues such as testes, liver, kidneys, and neuronal cells [29,30]. Among the properties attributed to EGCG is its antioxidant capacity, which consists of its ability to scavenge free radicals and chelate heavy metals [26,27,28].

The present study investigated the antioxidant effect of EGCG on Cd-induced toxicity when administered during the prepubertal stage, a phase in which reproductive organs, such as the testes and epididymides, undergo increased function and development, as reflected by an increase in serum testosterone concentrations and organ weight. Previous studies have reported that Cd induces alterations in the activity of antioxidant enzymes, such as SOD, CAT, and GPx, and histopathological changes in the testes of adolescent rats [18,24]. However, in our study, we observed an increase in Cd bioaccumulation, a decrease in testosterone concentration, and an increase in lipoperoxidation in the testes and the caput and cauda regions of the epididymides. Additionally, we noted alterations in the total activity of SOD and its isoforms Cu/Zn-SOD and Mn-SOD, as well as histological changes in the blood vessels and Leydig cells present in the seminiferous tubules. We also observed alterations in the epithelium of the cauda and caput regions of the epididymides. When EGCG was supplemented, a decrease in Cd bioaccumulation was observed, while testosterone concentrations remained unchanged, and a reduction in lipoperoxidation concentration was noted. The antioxidant enzyme activity, including SOD isoforms, was preserved at levels reached by the Ctrl group. In addition to protecting the architecture of the testicular epithelium and the caput and cauda regions of the epididymis, the administration of Cd+EGCG decreased Cd bioaccumulation, restored testosterone concentration, and fully restored antioxidant enzyme activity, including the activity of SOD isoforms. Several studies have described the bioaccumulation of Cd in the blood and various organs, including the testes [21,31,32]. It is known that the half-life of Cd in human blood is approximately 3 to 4 months, which is significantly shorter than its half-life in organs, which has been reported to be 10 to 30 years in humans and animals. Meanwhile, hemoglobin (Hb) is recognized as the main protein that binds to Cd within red blood cells [31,33,34]. In humans, especially teenagers, exposed to an environment with high Cd concentrations, this metal bioaccumulates in the blood [35]. Similarly, in pubertal rats exposed to Cd (1 mg/kg body weight) from PND 1 to 35 and 49, bioaccumulation of Cd was observed in blood, testes, and epididymides [18], which is consistent with our results. On the other hand, the ability of EGCG to scavenge free radicals may be related to the chelation of heavy metals by its phenolic groups [26,36], converting them into inactive molecules that are more soluble in water and should facilitate their excretion via the kidneys [23,26], helping to reduce the concentrations of Cd. The results of the present study suggest that the reduction in Cd concentration in the blood and reproductive organs of rats treated with Cd+EGCG may indicate that this catechin is able to bind to Cd actively. EGCG promotes the elimination of Cd from the blood by binding to this heavy metal through hydroxyl groups, thereby inactivating Cd through chelation. This process involves the -OH groups located on rings B and D, with the latter being the gallate group [25,37], reacting with the free radicals generated by Cd to form a stable ring-shaped structure called a cyclic chelate. This molecule increases solubility in water, facilitating its excretion in urine and eliminating Cd from the blood and tissues [25,38]. This Cd chelation mechanism is due to the presence of eight hydroxyl groups in the EGCG structure, located in positions 3′, 4′, and 5′, with a gallate fraction in C-3. This makes it a better electron donor and, therefore, a better free radical scavenger [28,39]. These findings suggest that EGCG may improve Cd excretion from the body, offering a potential strategy to reduce Cd burden. During prepuberty and puberty, some processes indicate the transition from childhood to adulthood in rodents, including development and growth in the testes [6] and the cellular differentiation and expansion of the epididymides [4]. In our study, daily exposure to 1 mg/kg body weight of CdCl_2_ for 28 days resulted in decreased weight of the testes and the caput and cauda of the epididymides. These results agree with those of Hernández-Rodríguez et al. [18] and Yang et al. [40], who administered CdCl_2_ to 5-week-old male mice at doses of 0.5, 1.5, and 2.5 mg/kg body weight and observed a decrease in testicular weight and testosterone concentration. This indicates that Cd impairs Leydig cell proliferation and testosterone synthesis. The testes and epididymides, as well as accessory organs, can serve as indicators of changes in androgens. According to Wen et al. [41], the population of immature Leydig cells decreases at PND 33 to allow the proliferation of mature Leydig cells during puberty, enabling the secretion of testosterone. In this way, the size and weight of these organs are regulated by testosterone, which also promotes cell differentiation [42,43,44]. Testosterone secretion is crucial for sustaining sperm production in adult males [45]. In our study, the maintenance of serum testosterone concentration with EGCG supplementation was similar to that of the Ctrl group. In contrast, Cd+EGCG was able to fully restore testosterone concentration compared to the Ctrl group. It has been reported that exposure to Cd can interfere with the activity of the HPG axis [46]. Studies suggest that Cd leads to a decrease in serum testosterone concentration [40,47] by inhibiting steroidogenesis via the enzymes cytochrome P540 and 17β-hydroxysteroid dehydrogenase, which participate in testosterone synthesis, as well as the StAR protein (steroidogenic regulatory protein) [48,49,50]. Some studies suggest that EGCG restores testosterone concentrations in models of testicular damage caused by the administration of cisplatin (10 mg/kg) [51] or by experimental cryptorchidism [52] in rats and rabbits, respectively.

Different studies have indicated that Cd toxicity is mediated by multiple mechanisms, including oxidative stress, inflammation, apoptosis, and endocrine disruption [24,33], independent of the timing and age of Cd exposure. Exposure to Cd in rats at 1, 3, 10, and 21 different PNDs resulted in a gradual decrease in the activities of SOD, CAT, and GTS (total glutathione transferase) in the testes, and an increase in MDA levels was also observed [20]. A similar effect was observed when Cd was administered to rats at PND 40 for 28 days, reducing SOD, CAT, and GPx activities and increasing lipoperoxidation [24]. Additionally, administration of Cd to rats from PND 1 to 49 increased SOD and CAT activity in the testes and seminal vesicle; however, a decrease was observed in the epididymis [18]. Lamas et al. [22] also found a decrease in SOD and CAT activity in the epididymis after a single administration of 1.2 mg/kg body weight CdCl_2_, i.p., in adult rats. Our results are consistent with those previously reported on antioxidant enzyme activity and the effects of Cd exposure, namely a decrease in SOD, CAT, and GPx activity in the testes and epididymides. The activity of total SOD is reflected in Cu/Zn-SOD and Mn-SOD isoforms; a decrease in their activity was observed in the present study. It has been reported that Cd displaces Mn, Zn, and Cu from the active site of SOD, as well as Fe from the heme group of CAT and Se from the GPx enzyme [23]. Shi and Fu [53] orally administered 5 mg/kg body weight of CdCl_2_ to rats for 28 days and observed inhibition of the expression of the nuclear factor erythroid 2-related factor 2 (Nrf2) and decreased enzyme levels related to redox balance in the testes and epididymides. This fact is an adaptive response to maintain cellular homeostasis. In the present study, EGCG in Cd+EGCG restored antioxidant activity, resulting in a reduction in lipoperoxidation in the reproductive organs examined. Consistent with this, a study by Singh et al. [54] found that administering As (20 mg/kg) to mice resulted in a reduction in Nrf_2_ expression. Research suggests that EGCG helps maintain redox balance by enhancing the intracellular antioxidant response system and increasing the expression of antioxidant genes such as SOD, CAT, and GPx [25]. This process occurs through the modulation of the Nrf2 signaling pathway [25,55,56]. Additionally, EGCG is recognized as an antioxidant that activates Nrf2 via the PI3K/Akt (phosphatidylinositol 3-kinase/protein kinase B) and ERK1/2 (extracellular signal-regulated protein kinase) signaling pathways [25,57]. Additionally, it has been suggested that EGCG maintains the redox balance by enhancing the intracellular antioxidant response system and increasing the expression of antioxidant genes (SOD, CAT, and GPx) through modulation of the Nrf2 signaling pathway [25,55,56]. EGCG has been identified as antioxidant that activates Nrf_2_ via the PI3K/Akt (phosphatidylinositol 3-kinase/protein kinase B) and ERK1/2 (extracellular signal-regulated protein kinase) signaling pathways [25,57].

During puberty, active spermiogenesis occurs in the presence of elongated spermatids and a continuous increase in the lumen diameter of the seminiferous tubules [24,58]. Cell differentiation and expansion also occur in the epididymis from PND 21 to 49 [4,59]. Cd in the testes damages the structure of the blood–testis barrier (BTB) and the vascular endothelium and leads to alterations of germ cells, Sertoli cells, and Leydig cells [21,60]. The epithelium of the epididymides is affected by Cd, resulting in a reduction in the tubular lumen [22,60]. Exposure to Cd can damage testicular histology and lead to disruption of cell adhesion and loss of germ cells, thereby impairing the process of spermatogenesis [19]. Additionally, alterations of the layers integrating the interstitial blood vessels were observed in mice treated with 2.5 mg/kg CdCl_2_ for 10 days [40]. This is important because the vascular endothelium is relevant for the endocrine function of the testis, and damage to the integrity of the BTB leads to inflammation and apoptosis in the testis [21]. Leite et al. [61] demonstrated in rats that a single injection of 1.5 mg/kg body weight CdCl_2_ resulted in a decrease in the lumen of the seminiferous tubules and a reduction in the number of Leydig cells. In the present study, histological analysis revealed that Cd affects the epithelium of seminiferous tubules, with changes including cell desquamation, basal membrane folding, and spermatocyte hypertrophy. Additionally, there are dilated blood vessels and atrophy of the Leydig cells in the interstitium. The increase in vasodilatation suggests that Cd alters the expression and function of vascular endothelial cadherin, a major component of endothelial cell junctions [21]. Cd has also been shown to change the stability and expression of adhesion proteins, such as occludins [29], leading to the disruption of connections between endothelial cells in capillaries and venules [60]. The alteration of cell adhesion proteins and the loss of germ cells result in the disruption of the epithelial architecture of the seminiferous tubules and the folding of the basal lamina.

Moreover, Cd has been shown to cause apoptosis in Leydig cells by promoting excessive mitochondrial fission and impairing mitophagy, leading to a reduction in Leydig cell numbers [24]. In addition to its bioaccumulation, Cd also affects inflammatory processes that can trigger a permanent loss of spermatogenesis [22,62]. In the present study, EGCG supplementation showed improvements in the epithelia and interstitial zones of the reproductive organs, as well as hyperplasia of the Leydig cells. In contrast, Cd+EGCG exhibited partial recovery from the histopathologic changes in the seminiferous epithelium and blood vessels in the interstitial zone. However, hyperplasia of Leydig cells persisted, accompanied by a slight thickening of the basement membrane. It is possible that the Leydig cells are not fully functional. A previous study that administered 0.5 and 1 mg/kg CdCl_2_ showed that Leydig cell development is impaired [63]. In rabbits with experimental cryptorchidism, a condition that produces oxidative stress, EGCG was able to restore the seminiferous epithelium [52]. Lamas et al. [19] found that the administration of grape juice concentrate, which contains polyphenols such as EGCG and acts as a Cd chelator, led to a reduction in genotoxicity and inflammatory processes in the testes. This, in turn, led to the preservation of testicular cytoarchitecture. Regarding the epididymides and the effects of Cd on this organ, Lamas et al. [22] investigated the impacts of a single dose of 1.2 mg/kg CdCl_2_. They observed an increase in the epididymal epithelium, a lack of adherence between the principal cells and the basal cells, an increase in dense vesicles in the cytoplasm in the caput and cauda epididymal cells, and the presence of sperm in the lumen of the epididymal tubule. These results are consistent with our observations in the Cd groups. The epididymides in both regions presented altered morphology, increased height and area of the principal cells, and cellular disorganization. This increase in height and area of the principal cells could be due to the presence of dense vesicles. The cellular disorganization observed in the epithelium of both regions of the epididymides could be due to a lack of cell adhesion, as proposed by Lamas et al. [22]. In our study, administration of Cd+EGCG showed partial recovery of epididymal histology. This could be due to EGCG’s chelating mechanism against Cd and its ability to increase antioxidant activity. EGCG is an antioxidant compound that can be used in therapeutic studies to promote health. It eliminates free radicals, chelates heavy metals such as Cd, and improves the antioxidant response system by increasing Nrf2 pathway expression. However, its low bioavailability and difficulty in achieving therapeutic concentrations in tissues are significant limitations [25].

## 4. Materials and Methods

### 4.1. Experimental Animals

The 28 male Wistar rats included in this study had a body weight of 36.7 ± 4.2 g at PND 21. The animals were obtained from the vivarium facilities of the División de Ciencias Biológicas y de la Salud at the Universidad Autónoma Metropolitana. The rats were placed in acrylic boxes, with seven subjects per box, and maintained in an inverted 12:12-h light–dark cycle at a constant temperature of 24 °C. The rats had free access to food (Harlan Laboratories, Indianapolis, IN, USA) and water. The animals were handled according to the Mexican Official Norm (NOM-062-ZOO-1999, reviewed in 2021) and in conjunction with the institutional ethics committee (Conducción ética de la investigación, la docencia y la difusión en la División de Ciencias Biológicas y de la Salud de la UAM-Iztapalapa) as well as the National Institutes of Health (NIH, 2011). Experiments were conducted with the minimum number of animals necessary and without causing suffering.

### 4.2. Treatments

Four experimental groups were formed (n = 7 in each group), consisting of a Ctrl group (100 µL saline); Cd (1 mg/kg/100 µL CdCl_2_, Sigma, Chemical Co., St. Louis, MO, USA); EGCG (10 mg/kg/100 µL (–)-Epigallocatechin-3-gallate, Item No. 70935, Cayman Chemical Co., Ann Arbor, MI, USA); and Cd+EGCG (1 mg/kg/100 µL CdCl_2_, followed by 10 mg/kg/100 µL), administered one hour after the dose of Cd. All treatments were administered daily intraperitoneally (i.p.) in the prepubertal stage, from PND 21 to 49. The administered doses of CdCl_2_ and EGCG were chosen based on previous studies [30,46,48,64,65]. The initiation of administration corresponded to the pubertal stage. At the end of the treatments, euthanasia was performed by decapitation.

### 4.3. Experimental Procedure

The rats were euthanized by decapitation under deep anesthesia (ketamine–xylazine: 150–154 mg/Kg) [66]. Peripheral blood samples were collected into metal-free vacutainer tubes for quantification of Cd concentration, and blood serum for testosterone was quantified according to the kit’s instructions. The testes and epididymides samples were dissected bilaterally and weighed; the epididymides were regionalized into caput and cauda. The samples were divided for subsequent biochemical analyses, with 100 mg portions taken from the left testes and epididymides for biochemical determinations and Cd quantification. All samples were stored at −70 °C; the right testes and epididymides were used for histological processing and analysis (Figure 7).

### 4.4. Biochemical Analysis

#### 4.4.1. Cadmium Determination in Blood, Testes, and Epididymides

Cd content in blood, testes, and epididymides was determined according to the protocol of Sharma et al. [67], using an atomic absorption spectrophotometer (AAS) (Perkin Elmer, PinAAcle Model AS900; Norwalk, CT, USA) with a graphite furnace (THGA) and a sampler (AS900, Perkin Elmer). For each analysis, calibration curves were established using aqueous Cd standards (0.5, 1.0, 2.0, 4.0, and 6.0 µg/L), which enabled the interpretation of Cd levels (GFAA Stock mixed standard, PerkinElmer, Mexico City, Mexico). Tissues and blood were digested, and approximately 100 mg and 200 µL, respectively, were collected in polypropylene tubes with the addition of a mix of nitric, perchloric, and sulfuric acids (5:2:1) (Suprapure, Merck, Kenilworth, Mexico City, Mexico). The liquid resulting from the acid digestion was analyzed using an AAS. To ensure quality control of Cd measurement, during every analytical session, a biological external reference material was measured (1577b, Bovine Liver, National Institute of Standards Technology) (NIST 1577b, USA) [102-NIST 1577b; Bovine Liver SRM 1577b. National Institute of Standards and Technology: Gaithersburg, MD, USA, 1991]. The concentration of Cd was expressed as μg of Cd/g tissue and μg of Cd/mL for blood.

#### 4.4.2. Quantification of Serum Testosterone

Blood samples were obtained by decapitating each animal between 13:00 and 14:00 h to avoid circadian variations and collected in tubes with serum separator stoppers (BD Vacutainer SST, Mexico City, Mexico). Serum was obtained by centrifugation for 15 min at 3000 rpm. Free testosterone was quantified by enzyme-linked immunosorbent assay (ELISA). Each sample was quantified in duplicate using a commercial kit (Testosterone ELISA EIA-1559, DRG, Springfield, NJ, USA). The kit was read using a UV-Vis spectrophotometer (Perkin-Elmer Lambda 40, Norwalk, CT, USA) at an optical density of 450 nm. The quantification of serum testosterone was expressed as ng/mL.

#### 4.4.3. Quantification of Malondialdehyde (MDA)

MDA was measured using the thiobarbituric acid reactive substances quantification technique. The testes and both regions of the epididymides were homogenized in Tris buffer solution (Tris-HCl, 150 mM, pH 7.4; Sigma-Aldrich, Co., St. Louis, MO, USA). The samples were then incubated in boiling water for 45 min. Following this, the samples were read at 532 nm using a UV-Vis spectrophotometer (PerkinElmer Lambda 40). The MDA concentration was calculated using an extinction coefficient of 1.56 × 10^5^ cm^−1^ M^−1^ and expressed as MDA (μmol/mg protein) [68]. The protein concentration was determined by the Bradford technique [69].

### 4.5. Determination of Antioxidant Enzyme Activity

#### 4.5.1. Superoxide Dismutase

Total SOD activity was determined according to the protocol of Schwartz et al. [70]. The testes and regions of the epididymides were homogenized in 20 mM carbonate buffer (Sigma-Aldrich, Co., St. Louis, MO, USA), pH 10.2, in 0.02% Triton X-100 (Sigma-Aldrich, Co., St. Louis, MO, USA). The samples were read at 550 nm using a UV-Vis spectrophotometer (Perkin Elmer, Lambda 40) every 30 s for 3 min. The difference between total SOD and Mn-SOD activity is the activity corresponding to Cu/Zn-SOD when inhibited with 1 mM potassium cyanide (Sigma-Aldrich, Co., St. Louis, MO, USA). The results were expressed as percentage variation against the respective Ctrl values of SOD (U/mg protein).

#### 4.5.2. Catalase

CAT enzyme activity was determined according to the protocol of Aebi [71]. The testes and regions of the epididymides were homogenized in 50 mM Tris-HCl buffer (pH 7.4). The samples were analyzed by UV-Vis spectrophotometry (Perkin Elmer, Lambda 40, Norwalk, CT, USA) at a wavelength of 240 nm every 1 min for 3 min. The enzyme activity was expressed as CAT (U/mg protein).

#### 4.5.3. Glutathione Peroxidase

GPx activity was measured according to the protocol of Hafeman et al. [72]. The testes and regions of the epididymides were homogenized in phosphate-EDTA buffer (PB-EDTA), 50 mM, pH 10, with 25% meta-phosphoric acid (Sigma-Aldrich, Co., St. Louis, MO, USA). The samples were analyzed by UV-Vis spectrophotometry (Perkin Elmer, Lambda 40, Norwalk, CT, USA) and measured at 412 nm for 2 min. The results were expressed as GPx (mmol/g protein).

### 4.6. Histological Analysis

Samples of testes and regions of the epididymides (caput and cauda) were fixed in Karnovsky’s solution (without Ca, pH 7.4) for 24 h [73]; then, they were postfixed in 1% osmium tetroxide (Sigma-Aldrich, Co., St. Louis, MO, USA) and dehydrated in a graded ethanol series for subsequent inclusion in EPON 812 resin (Ted Pella, INC., Redding, CA, USA). Semi-thin sections of 1 µm thickness were cut with an Ultracut UCT microtome (Leica, model Ultracut-UCT, Microsystems Wetzlar, Alemania) and stained with 0.5% toluidine blue. Histological analysis of the seminiferous tubules was performed using a light microscope (Olympus BX 51, Tokyo, Japan). The seminiferous epithelium maturation index or Johnsen index [74], fifteen cross sections of seminiferous tubules in stage VIII of the seminiferous epithelium cycle, and a homogeneous tubular structure per subject were analyzed. The area of the seminiferous epithelium was determined by the difference between the measurement of the tubule (external part) and the lumen measurement (internal part) using an image analysis system (Image-Pro Plus 5.1, Media Cybernetics, INC., Rockville, MD, USA). A score of 1 to 10 was assigned to each seminiferous tubule concerning the type of cell present—zero for no cell type and ten for complete spermatogenesis—as previously evaluated [75]. The histopathologic index [76] was determined by the evaluation of 15 cross sections of seminiferous tubules in stage VIII of the seminiferous epithelium cycle for each animal. A score ranging from 1 to 6 was assigned as follows: 1 for the presence of basal lamina folding and cell desquamation; 2 for epithelial vacuolization, multinucleated cells, and pyknosis; 3 for seminiferous tubules without spermatids; 4 for tubules without spermatocytes; 5 for tubules without spermatogonia; and 6 for the absence of all cell types [76]. In both regions of the epididymides, 15 transverse sections of the epididymal tubules per subject were analyzed, where the area and height of the principal cells were determined using an image analysis system (Image-Pro-Plus 5.1, Media Cybernetics, INC., MD, USA).

### 4.7. Statistical Analysis

Statistical analyses were performed using the GraphPad Prism software package (version 8.0.1). All data are expressed as mean ± standard error of the mean (SEM). The organ weights, Cd concentration, serum testosterone concentration, antioxidant enzyme activity, and histological analysis were statistically analyzed. The data that passed the normality and variability tests were subsequently analyzed using the one-way analysis of variance (ANOVA) parametric test, followed by a multiple comparison Tukey–Kramer test. Values obtained at *p* < 0.05 were considered statistically significant.

## 5. Conclusions

Our findings indicate that EGCG supplementation during Cd exposure in the prepubertal stage mitigates Cd-induced toxicity by modulating antioxidant activity and preserving the structural integrity of the testicular and epididymal epithelia. This could potentially promote sexual and reproductive function in maturity. EGCG exerts protective effects against Cd-induced reproductive damage during the prepubertal period, suggesting its potential therapeutic use to counteract Cd toxicity in reproductive development.

## Figures and Tables

**Figure 1 ijms-26-08264-f001:**
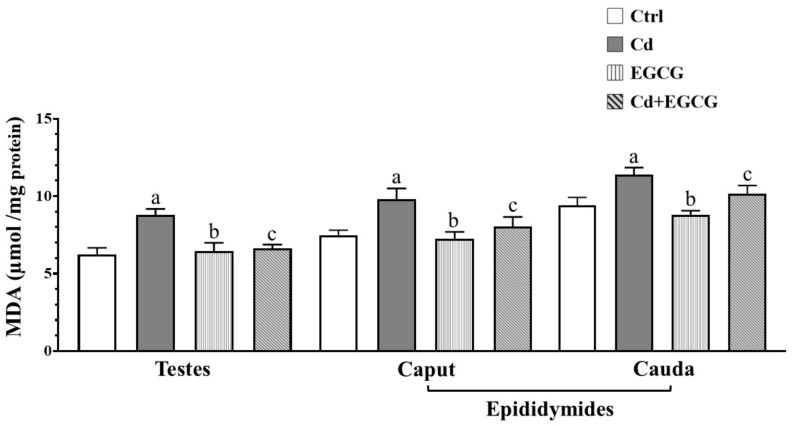
Determination of MDA in the reproductive organs for all treatment groups. The Cd showed a significant increase in MDA concentrations in the testes and epididymides regions. In contrast, the EGCG and Cd+EGCG showed a significant decrease in MDA concentrations in the testes, caput, and cauda epididymides regions. Data are presented as mean ± SEM for all analyzed groups (n = 7/treatment group). ^a^ indicates a significant difference (*p* < 0.05) between Ctrl and Cd; ^b^ indicates a significant difference (*p* < 0.05) between Cd and EGCG; ^c^ indicates a significant difference (*p* < 0.05) between Cd and Cd+EGCG. One-way ANOVA followed by the Tukey–Kramer test.

**Figure 2 ijms-26-08264-f002:**
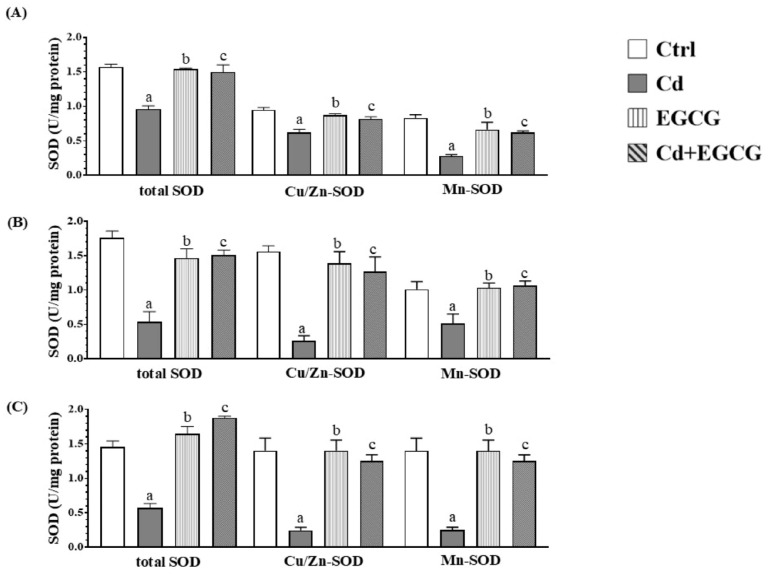
SOD activity in the reproductive organs for all treatment groups. The Cd showed a significant increase in SOD activity in the testes (**A**), caput (**B**), and cauda (**C**) regions of the epididymides. In contrast, EGCG and Cd+EGCG showed a significant decrease in SOD activities in the testes (**A**), caput (**B**), and cauda (**C**) of the regions. Data are presented as mean ± SEM for all analyzed groups (n = 7/treatment group). ^a^ indicates a significant difference (*p* < 0.05) between Ctrl and Cd; ^b^ indicates a significant difference (*p* < 0.05) between Cd and EGCG; ^c^ indicates a significant difference (*p* < 0.05) between Cd and Cd+EGCG. One-way ANOVA followed by the Tukey–Kramer test.

**Figure 3 ijms-26-08264-f003:**
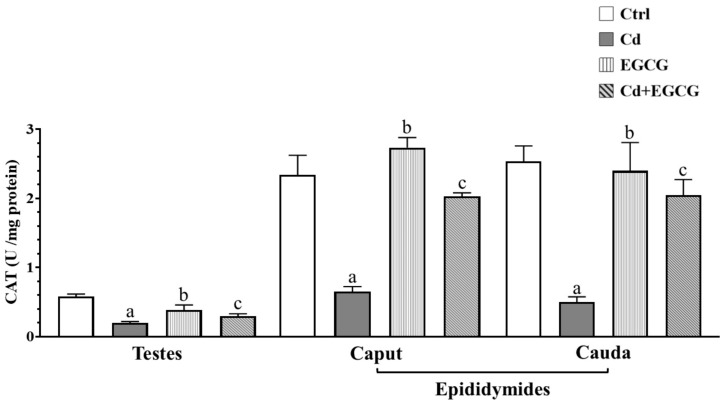
CAT activity in the reproductive organs. A significant decrease was observed in the testes and both epididymides regions in the Cd treatment groups. In contrast, the EGCG and Cd+EGCG groups showed a significant increase in the testes, caput, and cauda of the epididymides. Data are presented as mean ± SEM for all analyzed groups (n = 7/treatment group). ^a^ indicates a significant difference (*p* < 0.05) between Ctrl and Cd; ^b^ indicates a significant difference (*p* < 0.05) between Cd and EGCG; ^c^ indicates a significant difference (*p* < 0.05) between Cd and Cd+EGCG. One-way ANOVA followed by the Tukey–Kramer test.

**Figure 4 ijms-26-08264-f004:**
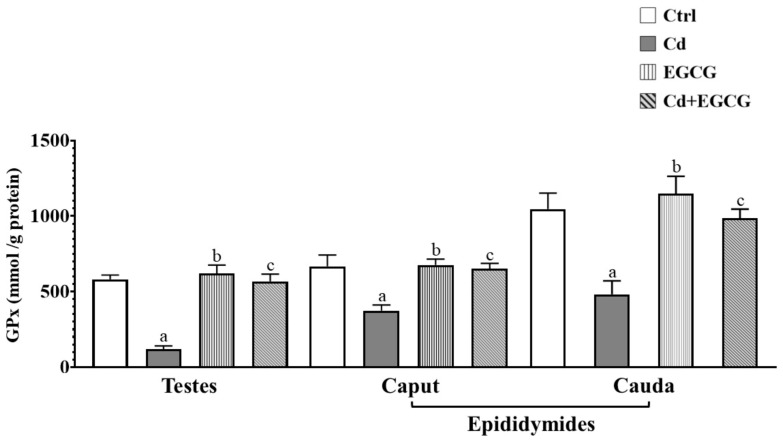
GPx activity in the reproductive organs. A significant decrease was observed in the Cd group in the testes and both regions of the epididymides. In contrast, the EGCG and Cd+EGCG groups demonstrated a significant increase in the testes and both regions of the epididymides. Data are presented as mean ± SEM for all analyzed groups (n = 7/treatment group). ^a^ indicates a significant difference (*p* < 0.05) between Ctrl and Cd; ^b^ indicates a significant difference (*p* < 0.05) between Cd and EGCG; ^c^ indicates a significant difference (*p* < 0.05) between Cd and Cd+EGCG. One-way ANOVA followed by the Tukey–Kramer test.

**Figure 5 ijms-26-08264-f005:**
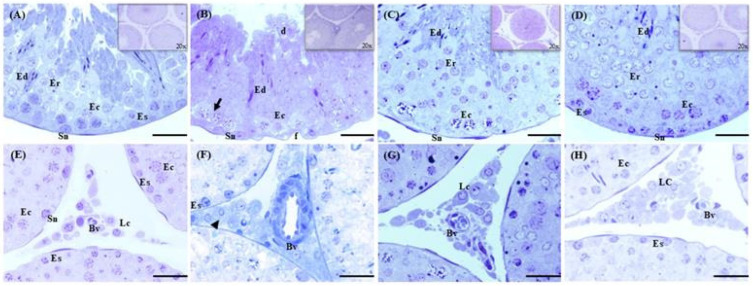
Testes photomicrograph. Ctrl (**A**), Cd (**B**), EGCG (**C**), and Cd+EGCG (**D**), along with the interstitial zone of the seminiferous tubules: Ctrl (**E**), Cd (**F**), EGCG (**G**), and Cd + EGCG (**H**). In the Ctrl group (**A**), Sertoli nucleus (Sn), as well as spermatocytes (Ec) and elongated spermatids (Ed), are present, with no alterations in the basement membrane; in the interstitial zone (**E**), blood vessels (BV) and Leydig cells (Lc) show no alterations. In the Cd group (**B**), basement membrane folding (f), desquamation (d), and spermatocyte hypertrophy (black arrow) are observed; regarding the interstitial zone, dilated blood vessels (Bv) are noted in the Cd group (**F**), along with atrophied Leydig cells (arrowhead). In the EGCG group (**C**), recovery of the basement membrane is noted without alterations in the germ cells, including spermatogonia (Es), spermatocytes (Ec), round spermatids (Er), elongated spermatids (Ed), and Sertoli nucleus (Sn), with no folding of the basement membrane; in the interstitial zone (**G**), blood vessels (BV) and Leydig cells (Lc) show no alterations. In the Cd+EGCG group (**D**), the Sertoli nucleus (Sn), spermatocytes (Ec), and elongated spermatids (Ed) are present without alterations in the basement membrane; in the interstitial zone (**H**), blood vessels (BV) and Leydig cells (Lc) are visible. Toluidine blue staining. 60X. Scale Bar 20 μm.

**Figure 6 ijms-26-08264-f006:**
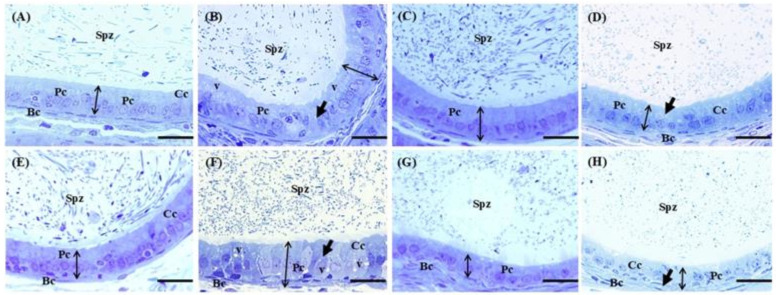
Epididymal photomicrograph. Caput region: Ctrl (**A**), Cd (**B**), EGCG (**C**), and Cd+EGCG (**D**). Cauda region: Ctrl (**E**), Cd (**F**), EGCG (**G**), and Cd+EGCG (**H**). Figures (**A**,**D**) show principal cells (Pc), clear cells (Cc), and basal cells (Bc). The presence of sperm in the lumen of the tubules in both regions is also evident. In the Cd group (**B**,**F**), cellular disorganization (black arrow) and vacuolization (v) are observed, as well as increased height of the principal cells, with a lower concentration of sperm in the lumen (Spz). In the EGCG treatment group (**C**,**G**), principal cells (Pc), clear cells (Cc), and basal cells (Bc) are shown, along with the presence of sperm in the lumen of the tubules. In the Cd+EGCG group (**D**,**H**), the presence of principal cells (Pc), clear cells (Cc), and basal cells (Bc) is shown, and sperm (Spz) in the lumen are visible. Samples were stained with toluidine blue, 60X. Scale bar 20 μm.

**Figure 7 ijms-26-08264-f007:**
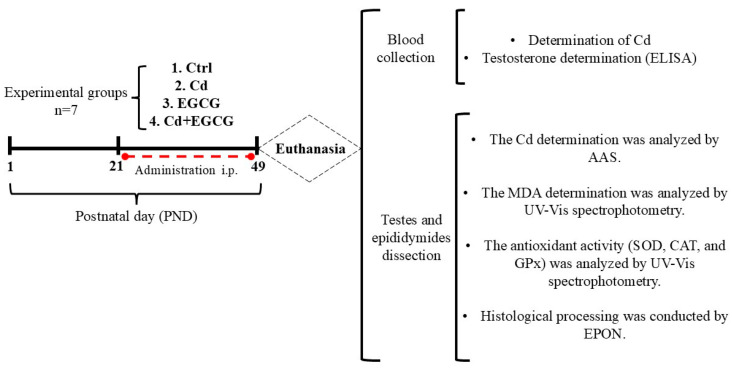
Experimental design and sample processing. Four experimental groups of Wistar rats were established: Ctrl (saline), Cd (CdCl_2_ 1 mg/kg body weight), EGCG (EGCG 10 mg/kg body weight), and Cd+EGCG (CdCl_2_+EGCG). The treatments were administered on the days indicated. At the end of treatment, on PND 49, the animals were euthanized to collect blood, testes, and epididymides samples. The following analyses were performed: Cd metal via AAS; testosterone concentration via ELISA; histological processing via EPON; and determination of the MDA concentration and antioxidant activity via UV-Vis spectrophotometry (SOD, CAT, and GPx). Postnatal day: PND, intraperitoneal: i.p., atomic absorption spectrophotometry: AAS, enzyme-linked immunosorbent assay: ELISA, epoxy resin: EPON, malondialdehyde: MDA, superoxide dismutase: SOD, catalase: CAT, glutathione peroxidase: GPx.

**Table 1 ijms-26-08264-t001:** Effects of Cd, EGCG, and Cd+EGCG on the weight and concentration of Cd in blood, testes, and epididymides.

	Ctrl	Cd	EGCG	Cd+EGCG
Testes weight (g)	1.40 ± 0.003	1.27 ± 0.016 ^a^	1.41 ± 0.035 ^b^	1.41 ± 0.029 ^c^
Testes Cd concentration (μg/g)	0.84 ± 0.19	4.77 ± 0.62 ^a^	0.59 ± 0.16 ^b^	0.62 ± 0.15 ^c^
Epididymides caput Cd concentration (μg/g)	0.86 ± 0.046	2.18 ± 0.32 ^a^	0.73 ± 0.14 ^b^	0.82 ± 0.16 ^c^
Caput weight (g)	0.085 ± 0.005	0.062 ± 0.004 ^a^	0.087 ± 0.005 ^b^	0.081 ± 0.003 ^c^
Epididymides cauda Cd concentration (μg/g)	0.64 ± 0.086	2.44 ± 0.16 ^a^	0.55 ± 0.12 ^b^	0.56 ± 0.12 ^c^
Cauda weight (g)	0.065 ± 0.005	0.039 ± 0.004 ^a^	0.071 ± 0.005 ^b^	0.068 ± 0.003 ^c^
Blood Cd concentration (μg/mL)	0.04 ± 0.011	0.23 ± 0.007 ^a^	0.04 ± 0.008 ^b^	0.04 ± 0.008 ^c^

Data are presented as mean ± SEM for all analyzed groups (n = 7/treatment group). ^a^ indicates a significant difference (*p* < 0.05) between Ctrl and Cd; ^b^ indicates a significant difference (*p* < 0.05) between Cd and EGCG; ^c^ indicates a significant difference (*p* < 0.05) between Cd and Cd+EGCG. One-way ANOVA followed by the Tukey–Kramer test.

**Table 2 ijms-26-08264-t002:** Serum testosterone concentration in the different treatments.

	Ctrl	Cd	EGCG	Cd+EGCG
ng/mL	2.39 ± 0.09	1.00 ± 0.07 ^a^	2.19 ± 0.21 ^b^	2.13 ± 0.21 ^c^

Data are expressed as means ± SEM for all analyzed groups (n = 7/treatment group). ^a^ indicates a significant difference (*p* < 0.05) between Ctrl and Cd; ^b^ indicates a significant difference (*p* < 0.05) between Cd and EGCG; ^c^ indicates a significant difference (*p* < 0.05) between Cd and Cd+EGCG. One-way ANOVA followed by the Tukey–Kramer test.

**Table 3 ijms-26-08264-t003:** Effects of Cd, EGCG, and Cd+EGCG on the measurement of histological parameters in the testes.

	Ctrl	Cd	EGCG	Cd+EGCG
Seminiferous epithelium area (µm^2^)	563.8 ± 10.7	474.9 ± 25.5 ^a^	572.9 ± 11.9 ^b^	521.5 ± 10.2 ^c^
Seminiferous epithelium diameter (µm)	329.4 ± 9.5	251.9 ± 3.7 ^a^	330.9 ± 11.2 ^b^	317.7 ± 11.4 ^c^
Histopathological index	1.53 ± 0.13	1.73 ± 0.11	1.46 ± 0.13	1.60 ± 0.13
Maturation index	8.6 ± 0.16	8.4 ± 0.12	8.7 ± 0.19	8.5 ± 0.18

Data are presented as mean ± SEM for all treatments (15 seminiferous tubules were evaluated for each subject and for each treatment). ^a^ indicates a significant difference (*p* < 0.05) between Ctrl and Cd; ^b^ indicates a significant difference (*p* < 0.05) between Cd and EGCG; ^c^ indicates a significant difference (*p* < 0.05) between Cd and Cd+EGCG. One-way ANOVA followed by the Tukey–Kramer test.

**Table 4 ijms-26-08264-t004:** Effects of Cd, EGCG, and EGCG+Cd on epididymal cell height and principal cell area of the two epididymal regions.

		Ctrl	Cd	EGCG	Cd+EGCG
Caput	Height (µm)	24.8 ± 0.68	33.2 ± 0.45 ^a^	24.6 ± 0.63 ^b^	23.4 ± 0.40 ^c^
	Area (µm^2^)	73.2 ± 1.87	81.0 ± 1.33 ^a^	73.2 ± 1.56 ^b^	73.6 ± 1.26 ^c^
Cauda	Height (µm)	23.4 ± 0.25	29.8 ± 0.48 ^a^	23.3 ± 0.24 ^b^	23.4 ± 0.40 ^c^
	Area (µm^2^)	64.4 ± 0.93	79.4 ± 1.78 ^a^	68.0 ± 1.72 ^b^	68.5 ± 1.95 ^c^

Data are presented as mean ± SEM for all treatments (15 epididymal tubules were evaluated for each subject and each treatment). ^a^ indicates a significant difference (*p* < 0.05) between Ctrl and Cd; ^b^ indicates a significant difference (*p* < 0.05) between Cd and EGCG; ^c^ indicates a significant difference (*p* < 0.05) between Cd and Cd+EGCG. One-way ANOVA followed by the Tukey–Kramer test.

## Data Availability

The original contributions presented in this study are included in the article. Further inquiries can be directed to the corresponding author.

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
