# Peer review of "Epigallocatechin-3-Gallate Enhances Antioxidant Activity and Improves Testicular and Epididymal Histology in Cadmium-Exposed Prepubertal Rats"

_ijms, 2025, doi:10.3390/ijms26178264_

Round 1
Reviewer 1 Report
Comments and Suggestions for Authors
Heavy mental is an essential environmental pollutant, and has deleterious influence on reproduction. How to improve the adverse impacts of heavy mental on reproduction is a big problem. Cd widely exists in water, soil, and fishes, which would be consumed by humans as food resulting to reproductive damages. In the present study, authors examined the influence of Cd on male reproductive organs, which can be alleviated by EGCG. Here is some comments to authors.
- Authors should supply more information on why EGCG was used in this study.
- EGCG is a antioxidant, but the Cd level in blood is decreased (table 1). How EGCG reduces the concentration of Cd in blood and reproductive organs?
- Authors showed that EGCG improved the reproductive function of male rats impaired by Cd through antioxidant, which is mediated by reducing the Cd level or increasing the activities of SOD and other enzymes?
- Line 113-115, authors analyzed the difference between Cd-exposed group and EGCG-supplement group. It is not necessary. Authors should analyze the statistical difference between EGCG-supplement and control, and Cd+EGCG-supplement and control.
- In section of 4.2, more information on how Cd and EGCG given, by intraperitoneal injection, subcutaneous injection, tail vein injection, or gavage, should be added.
Language should be further edited.
Author Response
Response to Reviewer
We thank the Editor and Reviewers for their careful reading of the manuscript in this review and their helpful suggestions for clarification and improvement.
Reviewer 1:
Comments and Suggestions for Authors
Heavy mental is an essential environmental pollutant, and has deleterious influence on reproduction. How to improve the adverse impacts of heavy mental on reproduction is a big problem. Cd widely exists in water, soil, and fishes, which would be consumed by humans as food resulting to reproductive damages. In the present study, authors examined the influence of Cd on male reproductive organs, which can be alleviated by EGCG. Here is some comments to authors.
- Authors should supply more information on why EGCG was used in this study.
We agree with the correction and were added in lines 93-106. The citations in parentheses are in the references section of the manuscript.
The paragraph: “EGCG has been reported to reduce the toxicity of heavy metals like nickel (Ni), arsenic (As), lead (Pb), mercury (Hg), and Cd in tissues of organs such as the testicles, liver, kidneys, and neuronal cells (23, 24). One of the properties attributed to EGCG is its antioxidant capacity, which involves eliminating ROS and chelating ionic metals (21, 25-26). Its high antioxidant activity depends on the presence of three hydroxyl groups (-OH) in its molecular structure (21). The antioxidant mechanism involves the donation of hydrogen atoms from the -OH groups to neutralize free radicals (23). Chelation occurs when these groups react with metal ions to form stable, ring-shaped structures called cyclic chelates. This structure increases the water solubility of EGCG, allowing it to be excreted in urine and eliminating heavy metals from tissues (21, 23, 24). Thus, the Cd that impacts reproductive processes could be mitigated by EGCG, which could favor reproductive health.
- EGCG is a antioxidant, but the Cd level in blood is decreased (table 1). How EGCG reduces the concentration of Cd in blood and reproductive organs?
Thank you for your comment. The specifications required were added to the line 322 to 328 and 336 to 346. In section 3. Discussion. The citations in parentheses are in the references section of the manuscript.
The paragraph: “It is known that the half-life of Cd in human blood is approximately 3 to 4 months, which is significantly shorter than its half-life in human organs, where the biological half-life of Cd in humans and animals has been reported to be 10 to 30 years, meanwhile, hemoglobin (Hb) is recognized as the main protein that binds to Cd within red blood cells (31, 32, 29). In humans, especially teenagers, exposed to an environment with high Cd concentrations, this metal bioaccumulates in the blood (33).”
“EGCG promotes the elimination of Cd from the blood by binding to this heavy metal through hydroxyl groups, thereby inactivating Cd through chelation. This process involves the -OH groups located on rings B and D, with the latter being the gallate group (23, 35), reacting with the free radicals generated by Cd to form a stable ring-shaped structure called a cyclic chelate. This molecule increases solubility in water, facilitating its excretion in urine and eliminating Cd from the blood and tissues (23, 36). This Cd chelation mechanism is due to the presence of eight hydroxyl groups in the EGCG structure, located in positions 3', 4', and 5', with a galate fraction in C-3. This makes it a better electron donor and, therefore, a better free radical scavenger (26, 37). These findings suggest that EGCG may improve Cd excretion from the body, offering a potential strategy to reduce Cd burden.”
- Authors showed that EGCG improved the reproductive function of male rats impaired by Cd through antioxidant, which is mediated by reducing the Cd level or increasing the activities of SOD and other enzymes?
Thank you for your comment. The specifications required were added to the line 395 to 402. In section 3. Discussion.The EGCG improves reproductive function through both mechanisms, by neutralizing free radicals and reactive oxygen species (ROS) and by chelating Cd. The citations in parentheses are in the references section of the manuscript.
The paragraph: “Additionally, it has been suggested that EGCG maintains the redox balance by enhancing the intracellular antioxidant response system and increasing the expression of antioxidant genes (SOD, CAT, and GPx) through modulation of the Nrf2 signaling pathway. (23, 54, 55). By activating the Nrf2 protein through the PI3K/Akt pathway (phosphatidylinositol 3-kinase and ERK1/2 signaling (extracellular signal-regulated protein kinase) signaling pathway, EGCG has been identified as a natural product that activates Nrf2 via the PI3K/Akt (phosphatidylinositol 3-kinase/protein kinase B) and ERK1/2 signaling pathway (23, 56).”
- Line 113-115, authors analyzed the difference between Cd-exposed group and EGCG-supplement group. It is not necessary. Authors should analyze the statistical difference between EGCG-supplement and control, and Cd+EGCG-supplement and control.
Thank you for your comment. We reviewed the differences between the Ctrl and experimental groups. It has been corrected and modified in the tables and figures.
- In section of 4.2, more information on how Cd and EGCG given, by intraperitoneal injection, subcutaneous injection, tail vein injection, or gavage, should be added.
Thank you for your comment. The specifications required were added to the line 478 to 479.
The paragraph added:“All treatments were administered daily intraperitoneal via (i.p.) in the prepuberal stage, from 21 to 49 PND”
Reviewer 2 Report
Comments and Suggestions for Authors
The article entitled: " Epigallocatechin-3-gallate enhances the antioxidant activity and improves testicular and epididymal histology in cadmium-exposed prepubertal rats". The study's findings are valuable, however, in each section, several issues need to be significantly addressed.
Kindly revise the manuscript to reduce the similarity index and ensure proper paraphrasing and citation where necessary, to maintain academic integrity and avoid potential plagiarism concerns.
Abstract
- Please write the type of Epigallocatechin-3-gallate in the abstract section.
- Add recommendations and clinical implications in conclusion.
Introduction
- Add in-text citations forthe first two paragraphs.
- Mention sources of Cd exposure.
- Add the cadmium standard Permissible Exposure Limit (PEL) and mention high-risk populations.
- Expand previous studies on Cd exposure, toxic effects on histopathological and endocrine dysregulation.
- Mention the mechanism of action of EGCG and other studies that explored its antioxidant activity in testicular damage.
- The research gap and rationale are not mentioned. Please highlight them.
Methods
- Mention the age of the rat in the method section.
- Add reference of anesthesia (ketamine-xylazine: 150-4 mg/Kg.
- Reference of EGCG inappropriately cited as mentioned in the treatment section (The administered doses of CdCl2 and EGCG were chosen based on previous studies (28,41). Hence that both references were for Cd toxicity.
- Add Effects of Cd-exposed, EGCG-supplemented, and Cd+EGCG on the weight section.
- In Cadmium determination, please mention Cd interpretation levels.
- Mention the unit of Quantification of Serum Testosterone.
- Measuring testosterone only provides some information but is not enough for a complete assessment of testicular function, especially in the context of Cd exposure, which can affect multiple components of the hypothalamic-pituitary-gonadal (HPG) axis.
- The seminiferous epithelium maturation index or Johnsen index, and the histopathologic index are not well clarified. Please mention in brief its interpretation.
Results
- In the Effects of Cd-exposed, EGCG-supplemented, and Cd+EGCG on the weight section please identify if the difference is increased or reduced significance.
Discussion
- It is mentioned that (It is the most active compound (24) that was used to reduce the toxicity of heavy metals such as nickel (Ni), arsenic (As), lead (Pb), ,,,,. Please add that it is one of the most.
- Discuss how EGCG leads to a decrease in Cd bioaccumulation.
- Mention the study's strengths and limitations.
Conclusion
Please add recommendations and clinical implications in the conclusion section
Author Response
Response to Reviewer 2
We thank the Editor and Reviewers for their careful reading of the manuscript in this review and their helpful suggestions for clarification and improvement.
Comments and Suggestions for Authors
The article entitled: " Epigallocatechin-3-gallate enhances the antioxidant activity and improves testicular and epididymal histology in cadmium-exposed prepubertal rats". The study's findings are valuable, however, in each section, several issues need to be significantly addressed.
Kindly revise the manuscript to reduce the similarity index and ensure proper paraphrasing and citation where necessary, to maintain academic integrity and avoid potential plagiarism concerns.
We appreciate your comments. We have verified that we comply with the requirements for writing our own paragraphs and backing up the information with correctly placed citations to avoid unnecessary situations of plagiarism.
Abstract
- Please write the type of Epigallocatechin-3-gallate in the abstract section.
Thank you for your comments, (–)-Epigallocatechin-3-gallate (EGCG) was used in this work and was acquired with Item No. 70935. Cayman Chemical Co. Ann Harbor, MI, USA. This information was added in the line 34.
- Add recommendations and clinical implications in conclusion.
Thank you for your recommendation. We believe that this is more of a therapeutic solution than a clinical one. That is why we added the following paragraph "These findings demonstrate that EGCG exerts protective effects against Cd-induced reproductive damage during the prepuberal period, suggesting its potential therapeutic use to counteract Cd toxicity in the reproductive development” in the lines 46 to 48.
Introduction
- Add in-text citations for the first two paragraphs.
We appreciate your observation. The references were added where requested.
Spaziani, M., Tarantino, C., Tahani, N., Gianfrilli, D., Sbardella, E., Lenzi, A., & Radicioni, A. F. (2021). Hypothalamo-Pituitary axis and puberty. Molecular and cellular endocrinology, 520, 111094. https://doi.org/10.1016/j.mce.2020.111094.
Castellano JM, Tena-Sempere M. Kisspeptins and Puberty. Rev Esp Endocrinol Pediatr.2017;8(2):8–14.
- Mention sources of Cd exposure.
Thank you for your recommendations. This information was added in the introduction section in lines 70 to 77.
The paragraph added: “This heavy metal primarily results from human activities, with mining and the metallurgical, textile, and automotive industries being the main sources. In addition, the use of pesticides, insecticides, and fertilizers contaminates food and drinking water. According to the Occupational Safety and Health Administration (OSHA) of the United States, the permissible exposure limit (PEL) for occupational exposure to Cd or Cd oxide is 0.1 mg/m³ in a workday (9).”
Add the cadmium standard Permissible Exposure Limit (PEL) and mention high-risk populations.
Thank you for your recommendations. This information was added in the introduction section in lines 73 to 77. The citations in parentheses are in the references section of the manuscript.
The paragraph: “According to the Occupational Safety and Health Administration (OSHA) of the United States, the permissible exposure limit (PEL) for occupational exposure to Cd or Cd oxide is 0.1 mg/m³ in a workday (9). Meanwhile, the World Health Organization (WHO, 2019) establishes that the tolerable monthly ingestion of Cd is 25 µg/kg.”
- Expand previous studies on Cd exposure, toxic effects on histopathological and endocrine dysregulation.
Thank you for your recommendations. This information was added in the introduction section in lines 85 to 89. The citations in parentheses are in the references section of the manuscript.
The paragraph added: “Cd causes damage to the structure of the testicular and epididymal epithelium, such as cellular disorganization and epithelial desquamation, loss of germ cells, tubular degeneration with vacuoles, apoptosis of spermatogenic cells (17, 22), lack of adhesion between principal cells and basal cells, and a reduction in testosterone concentration (22).”.
- Mention the mechanism of action of EGCG and other studies that explored its antioxidant activity in testicular damage.
Thank you for your recommendations. This information was added in the introduction section in lines 93 to104. The citations in parentheses are in the references section of the manuscript.
The paragraph added: “EGCG has been reported to reduce the toxicity of heavy metals like nickel (Ni), arsenic (As), lead (Pb), mercury (Hg), and Cd in tissues of organs such as the testicles, liver, kidneys, and neuronal cells (23, 24). One of the properties attributed to EGCG is its antioxidant capacity, which involves eliminating ROS and chelating ionic metals (21, 25-26). Its high antioxidant activity depends on the presence of three hydroxyl groups (-OH) in its molecular structure (21). The antioxidant mechanism involves the donation of hydrogen atoms from the -OH groups to neutralize free radicals (23). Chelation occurs when these groups react with metal ions to form stable, ring-shaped structures called cyclic chelates. This structure increases the water solubility of EGCG, allowing it to be excreted in urine and eliminating heavy metals from tissues (21, 23, 24). Thus, the Cd that impacts reproductive processes could be mitigated by EGCG, which could favor reproductive health”
- The research gap and rationale are not mentioned. Please highlight them.
Thank you for your recommendations. This information was added in the Introduction section in lines 104 to 106.
The paragraph added: “In the present study, the effect of EGCG on antioxidant activity and the improvement of testicular and epididymal histology in cadmium-exposed prepubertal rats was evaluated.”
Methods
- Mention the age of the rat in the method section.
Thank you for your comment. The information was added in 4.1 Experimental animals section in line 462. Which was at 21 days postnatal (PND).
- Add reference of anesthesia (ketamine-xylazine: 150-4 mg/Kg.
Thank you for your comment. The reference was added in line 479, in the 4.3 Experimental procedure section.
Sotoudeh and Namavar, 2022: Sotoudeh N, Namavar MR. Optimisation of ketamine-xylazine anaesthetic dose and its association with changes in the dendritic spine of CA1 hippocampus in the young and old male and female Wistar rats. Vet Med Sci. 2022 Nov;8(6):2545-2552. doi: 10.1002/vms3.936. Epub 2022 Sep 12. PMID: 36097263; PMCID: PMC9677370.
- Reference of EGCG inappropriately cited as mentioned in the treatment section (The administered doses of CdCl2 and EGCG were chosen based on previous studies (28,41). Hence that both references were for Cd toxicity.
Thank you for your comment. We apologize for the error. We have added the correct references in line 481.
Ji YL, Wang H, Liu P, Wang Q, Zhao XF, Meng XH, Yu T, Zhang H, Zhang C, Zhang Y, Xu DX. Pubertal cadmium exposure impairs testicular development and spermatogenesis via disrupting testicular testosterone synthesis in adult mice. Reproductive Toxicology. 2010 Apr;29(2):176-83. Epub 2009 Nov 6. PMID: 19897027. doi:10.1016/j.reprotox.2009.10.014.
Lafuente A. The hypothalamic-pituitary-gonadal axis is target of cadmium toxicity. An update of recent studies and potential therapeutic approaches. Food and chemical toxicology : an international journal published for the British Industrial Biological Research Association. 2013;59:395-404.
Bernhoft RA. Cadmium toxicity and treatment. TheScientificWorldJournal. 2013;2013:394652; Ji YL, Wang H, Liu P, Wang Q, Zhao XF, Meng XH, et al. Pubertal cadmium exposure impairs testicular development and spermatogenesis via disrupting testicular testosterone synthesis in adult mice. Reproductive toxicology (Elmsford, NY). 2010;29(2):176-83.
García-Rodríguez Mdel C, Montaño-Rodríguez AR, Altamirano-Lozano MA. Modulation of hexavalent chromium-induced genotoxic damage in peripheral blood of mice by epigallocatechin-3-gallate (EGCG) and its relationship to the apoptotic activity. Journal of toxicology and environmental health Part A. 2016;79(1):28-38).
Yu NH, Pei H, Huang YP, Li YF. (-)-Epigallocatechin-3-Gallate Inhibits Arsenic-Induced Inflammation and Apoptosis through Suppression of Oxidative Stress in Mice. Cell Physiol Biochem. 2017;41(5):1788-1800. doi: 10.1159/000471911. Epub 2017 Apr 3. PMID: 28365685.
- Add Effects of Cd-exposed, EGCG-supplemented, and Cd+EGCG on the weight section.
Thank you for your comments. This question can be found in the 2. Results section, where we will provide an answer.
- In Cadmium determination, please mention Cd interpretation levels
Thank you for your comments. The information was added in 4.4.1. Cadmium determination in blood, testes, and epididymides section in line 510 to 512.
The paragraph: “For each analysis, calibration curves were established using aqueous Cd standards (0.5, 1.0, 2.0, 4.0, and 6.0 µg/L), which enabled the interpretation of Cd levels (GFAA Stock mixed standard, PerkinElmer, Mexico City, Mexico).”
- Mention the unit of Quantification of Serum Testosterone.
We thank you for your comment. It has been corrected and modified in the 4.4 of the Biochemical analysis section of serum testosterone quantification, on line 530.
The paragraph added: “The quantification of serum testosterone was expressed as ng/mL.”
- Measuring testosterone only provides some information but is not enough for a complete assessment of testicular function, especially in the context of Cd exposure, which can affect multiple components of the hypothalamic-pituitary-gonadal (HPG) axis.
Thank you for your feedback. We agree that testicular function is not only measured through testosterone quantification, which is why this study analyzed the histological characteristics of the testicular epithelium, such as the effects of FSH and LH gonadotropins. It has been reported that exposure to Cd can interfere with the activity of the HPG axis (Lafuente A. The hypothalamic-pituitary-gonadal axis is target of cadmium toxicity. An update of recent studies and potential therapeutic approaches. Food Chem Toxicol. 2013 Sep;59:395-404. doi: 10.1016/j.fct.2013.06.024. Epub 2013 Jun 26. PMID: 23811532).
- The seminiferous epithelium maturation index or Johnsen index, and the histopathologic index are not well clarified. Please mention in brief its interpretation.
Thank you for your review and comments. This information has been added to 4.6 Histological analysis section in the lines 574 to 588. The citations in parentheses are in the references section of the manuscript.
The paragraph added: “The seminiferous epithelium maturation index or Johnsen index (73), fifteen cross-sections of seminiferous tubules in stage VIII of the seminiferous epithelium cycle, and presenting a homogeneous tubular structure per subject, were analyzed. The area of the seminiferous epithelium was determined by the difference between the measurement of the tubule (external part) and the lumen measurement (internal part) using an image analysis system (Image-Pro Plus 5.1, Media Cybernetics, INC., MD, USA. A score of 1 to 10 was assigned to each seminiferous tubule concerning the type of cell present, zero for no cell type, and ten for complete spermatogenesis, as previously evaluated (74). The histopathologic index (75) was determined by the evaluation of 15 cross sections of seminiferous tubules in stage VIII of the seminiferous epithelium cycle for each animal. A score ranging from 1 to 6 was assigned as follows: 1 for the presence of basal lamina folding, cell desquamation; 2 for epithelial vacuolization, multinucleated cells, and pyknosis; 3 for seminiferous tubules without spermatids; 4 for tubules without spermatocytes; 5 for tubules without spermatogonia; and 6 for the absence of all cell types (75).”
Results
- In the Effects of Cd-exposed, EGCG-supplemented, and Cd+EGCG on the weight section please identify if the difference is increased or reduced significance.
We thank you for your comment. In Table 1, which refers to organ weights and cadmium concentration, we added: “Cd concentration below each reproductive organ” for greater clarity of the data shown. In addition, in the text in section 2. Results, in 2.1. Effects of Cd, EGCG, and Cd+EGCG on the weight and the concentration of Cd in blood, testes, and epididymides. Add "A significant decrease was found among the Cd groups compared to the Ctrl, EGCG, and Cd+EGCG groups (p < 0.05), in the lines 118 to 121.
Discussion
- It is mentioned that (It is the most active compound (24) that was used to reduce the toxicity of heavy metals such as nickel (Ni), arsenic (As), lead (Pb), ,,,,. Please add that it is one of the most.
We thank you for your comment. We apologize for the error, and added EGCG in line 296.
- Discuss how EGCG leads to a decrease in Cd bioaccumulation.
Thank you for your review and comments. This information has been added in lines 336 to 346. The citations in parentheses are in the references section of the manuscript.
The paragraph added: “EGCG promotes the elimination of Cd from the blood by binding to this heavy metal through hydroxyl groups, thereby inactivating Cd through chelation. This process involves the -OH groups located on rings B and D, with the latter being the gallate group (23, 35), reacting with the free radicals generated by Cd to form a stable ring-shaped structure called a cyclic chelate. This molecule increases solubility in water, facilitating its excretion in urine and eliminating Cd from the blood and tissues (23, 36). This Cd chelation mechanism is due to the presence of eight hydroxyl groups in the EGCG structure, located in positions 3', 4', and 5', with a galate fraction in C-3. This makes it a better electron donor and, therefore, a better free radical scavenger (26, 37). These findings suggest that EGCG may improve Cd excretion from the body, offering a potential strategy to reduce Cd burden.”
- Mention the study's strengths and limitations.
Thank you for your review and comments. “EGCG is an antioxidant compound that can be used in therapeutic studies to promote health. It eliminates free radicals, chelates heavy metals such as cadmium (Cd) and improves the antioxidant response system by increasing Nrf2 pathway expression. However, its low bioavailability and difficulty in achieving therapeutic concentrations in tissues are significant limitations (23). ”On lines 455 to 460. The citations in parentheses are in the references section of the manuscript
Conclusion
Please add recommendations and clinical implications in the conclusion section
Thank you for your recommendation. We believe that this is more of a therapeutic solution than a clinical one. That is why we added the following paragraph "These findings demonstrate that EGCG exerts protective effects against Cd-induced reproductive damage during the prepuberal period, suggesting its potential therapeutic use to counteract Cd toxicity in the reproductive development” in the lines 604 to 606.

Reviewer 3 Report
Comments and Suggestions for Authors The manuscript is well-structured, investigating the protective effects of Epigallocatechin-3-gallate (EGCG) against cadmium (Cd)-induced toxicity in the testes and epididymides of prepubertal male Wistar rats. The study is significant, exploring the impact of environmental toxicants on reproductive health during a critical developmental stage and evaluating the potential mitigative role of a natural antioxidant. The experimental design is rigorous, with clear methodology and appropriate statistical analysis. However, overall, the presentation of results is relatively superficial, lacking in-depth data analysis.- PleaseprovideadetailedexplanationofthescoringcriteriafortheJohnsenindexandthehistopathologicalindex.
- Theuseofone-wayanalysisofvariance(ANOVA)followedbytheTukey-Kramertestisappropriate.Itshouldbeclearlystatedwhetherallanalyzeddatameettheassumptionsofnormalityandhomogeneityofvariance.
- Itisrecommendedtoboldthelinesinthebargraphsthroughoutthemanuscript,standardizethefontsizeinimages,andaddressthebarelyvisiblesignificancelines.
- The results in the manuscript lack correlation analysis. Testosterone is primarily synthesized and secreted by testicular interstitial cells, yet this key result is not explored in depth in the discussion. References can be made to studies such as: Gut microbiota dysbiosis and oxidative damage in high-fat diet-induced impairment of spermatogenesis: Role of protocatechuic acid intervention (https://doi.org/10.1002/fft2.484)andPatternsofalterationinboarsemenqualityfrom9to37monthsoldandimprovementbyprotocatechuicacid(10.1186/s40104-024-01031-6).
- Evenwithintheseminiferousepitheliumofasingletestis,spermatogenesisoccursatdifferentstages.Resultsbasedonareacalculationsmayinvolvesomedegreeofrandomness.Itissuggestedtocategorizeandquantifybasedonpathologicalimagesoftheseminiferoustubules.
- Although the manuscript provides a relatively systematic interpretation with substantial pathological evidence, the results are entirely descriptive. It is recommended to supplement the discussion with content on the molecular mechanisms of EGCG, particularly regarding the regulation of antioxidant enzyme proteins and gene expression.
Author Response
Response to Reviewer 3.
We thank the Editor and Reviewers for their careful reading of the manuscript in this review and their helpful suggestions for clarification and improvement.
The manuscript is well-structured, investigating the protective effects of Epigallocatechin-3-gallate (EGCG) against cadmium (Cd)-induced toxicity in the testes and epididymides of prepubertal male Wistar rats. The study is significant, exploring the impact of environmental toxicants on reproductive health during a critical developmental stage and evaluating the potential mitigative role of a natural antioxidant. The experimental design is rigorous, with clear methodology and appropriate statistical analysis. However, overall, the presentation of results is relatively superficial, lacking in-depth data analysis.
We thank the Editor and Reviewers for their careful reading of the manuscript in this review and their helpful suggestions for clarification and improvement.
- Please provide a detailed explanation of thes coring criteria for the Johnsen index and the histopathological index.
Thank you for your review and comments. This information has been added to section 4.6 Histological analysis section in the lines 574 to 588. The citations in parentheses are in the references section of the manuscript.
The paragraph added: “The seminiferous epithelium maturation index or Johnsen index (73), fifteen cross-sections of seminiferous tubules in stage VIII of the seminiferous epithelium cycle, and presenting a homogeneous tubular structure per subject, were analyzed. The area of the seminiferous epithelium was determined by the difference between the measurement of the tubule (external part) and the lumen measurement (internal part) using an image analysis system (Image-Pro Plus 5.1, Media Cybernetics, INC., MD, USA. A score of 1 to 10 was assigned to each seminiferous tubule concerning the type of cell present, zero for no cell type, and ten for complete spermatogenesis, as previously evaluated (74). The histopathologic index (75) was determined by the evaluation of 15 cross sections of seminiferous tubules in stage VIII of the seminiferous epithelium cycle for each animal. A score ranging from 1 to 6 was assigned as follows: 1 for the presence of basal lamina folding, cell desquamation; 2 for epithelial vacuolization, multinucleated cells, and pyknosis; 3 for seminiferous tubules without spermatids; 4 for tubules without spermatocytes; 5 for tubules without spermatogonia; and 6 for the absence of all cell types (75)”
- Theuse of one-way analysis of variance (ANOVA) followed by the Tukey-Kramer testis appropriate. Its hould be clearly stated whetherallanalyzed data meettheas sumptions of normality and homogeneity of variance.
Thank you for your review and comments. This information has been added to section 4.7 of Statistical Analysis, on lines 594 to 599, which indicates that all data were analyzed and comply with normality and homogeneity of variance.
The paragraph added:“The organ weights, Cd concentration, serum testosterone concentration, antioxidant enzyme activity, and histological analysis were statistically analyzed. The data that passed the normality and variability tests were subsequently analyzed using the one-way analysis of variance (ANOVA) parametric test, followed by a multiple comparison Tukey-Kramer test. Values obtained at p < 0.05 were considered statistically significant.”
- It is recommended to bold the lines in the bargraphs through out the manuscript, standardize the font size in images, and address the barely visible significan celines.
Thank you for your review and comments. The requested changes have been made to the design of the graphs with regard to standardizing and normalizing the size and font throughout the manuscript.
The results in the manuscript lack correlation analysis. Testosterone is primarily synthesized and secreted by testicular interstitial cells, yet this key result is not explored in depth in the discussion. References can be made to studies such as: Gut microbiota dysbiosis and oxidative damage in high-fat diet-induced impairment of spermatogenesis: Role of protocatechuic acid intervention (https://doi.org/10.1002/fft2.484) and Patterns of alterationinboarsemenqualityfrom9to37monthsoldandimprovementbyprotocatechuicacid(10.1186/s40104-024-01031-6).
Thank you for your recommendation, this information was added in the section Discussion in the lines 359 to 360.
The paragraph added: “Testosterone secretion is crucial for sustaining sperm production in adult males (Hu R, Yang X, Gong J, Lv J, Yuan X, Shi M, Fu C, Tan B, Fan Z, Chen L, Zhang H, He J, Wu S. Patterns of alteration in boar semen quality from 9 to 37 months old and improvement by protocatechuic acid. J Anim Sci Biotechnol. 2024 May 17;15(1):78. doi: 10.1186/s40104-024-01031-6. PMID: 38755656; PMCID: PMC11100174.)”
- Even within these miniferous epithelium of a single testis, spermatogenesis occurs at different stages. Results base don area calculations may involve some degree of randomness. It is suggested to categorize and quantify based on pathological images of these miniferous tubules.
We appreciate your comments to improve our manuscript, which is why we included a detailed description to categorize the area and diameter of the seminiferous tubules.
This is described in the Materials and Metodhs section in 4.6 Histological analysis. In lines 574-580. The citations in parentheses are in the references section of the manuscript.
The paragraph added: “The seminiferous epithelium maturation index or Johnsen index (73), fifteen cross-sections of seminiferous tubules in stage VIII of the seminiferous epithelium cycle, and presenting a homogeneous tubular structure per subject, were analyzed. The area of the seminiferous epithelium was determined by the difference between the measurement of the tubule (external part) and the lumen measurement (internal part) using an image analysis system (Image-Pro Plus 5.1, Media Cybernetics, INC., MD, USA.”
- Although the manuscript provides a relatively systematic interpretation with substantial pathological evidence, the results are entirely descriptive. It is recommended to supplement the discussion with content on the molecular mechanisms of EGCG, particularly regarding the regulation of antioxidant enzyme proteins and gene expression.
Thank you for your comments. The information was added in 3. Discussion in lines 395 to 400. The citations in parentheses are in the references section of the manuscript.
The paragraph added: “it has been suggested that EGCG maintains the redox balance by enhancing the intracellular antioxidant response system and increasing the expression of antioxidant genes (SOD, CAT, and GPx) through modulation of the Nrf2 signaling pathway. (23, 54, 55). By activating the Nrf2 protein through the PI3K/Akt pathway (phosphatidylinositol 3-kinase and ERK1/2 signaling (extracellular signal-regulated protein kinase) signaling pathway, EGCG has been identified as a natural product that activates Nrf2 via the PI3K/Akt (phosphatidylinositol 3-kinase/protein kinase B) and ERK1/2 signaling pathway (23, 56).”

Reviewer 4 Report
Comments and Suggestions for Authors
Major points
1. The data indicated that the EGCG completely reduced Cd in the blood and tissues. What would be the possible mechanisms behind? Are there any previous studies on similar effects of EGCG or green tea's polyphenols? Are there any previous investigations indicating the detoxification of heavy metals by EGCG?
2. Since EGCG reduced the body's Cd concentration to the control level, all other data regarding the testis and epididymis would not make much sense. There would be no effects because of no Cd in the circulation. The authors should explore the reduction/detoxification of Cd by ECGC.
3. This study aims to investigate the mitigating effects of EGCG on the toxicity of Cd. Then the Introduction should provide the background of both the toxicity of Cd and the effects of EGCG. Please provide more information on previous studies of the toxicities of Cd and the bioactivities of polyphenols. Then, clearly state the aims of this study. Otherwise, all these descriptions regarding the prepubertal development of the male reproductive system are rather pointless. Please rewrite the Introduction accordingly.
4. Taking account of the two fundamental points above, the Discussion needs to be revised.
Minor points
1. Please change the group names to Control, Cd, EGCG, and Cd+EGCG in the entire text, figures, and tables.
2. Line 431, what does 'Animals that did not meet this criterion were excluded' mean? Does any group consist of fewer than 7 rats because 'not meeting this criteria'? Otherwise, please delete this sentence.
3. Figure 7, and Lines 447-451, were the chemicals injected on PND 49? If so, when (how long after) were the animals killed after the final injection?
Author Response
Response to Reviewer 4.
We thank the Editor and Reviewers for their careful reading of the manuscript in this review and their helpful suggestions for clarification and improvement.
Major points
The data indicated that the EGCG completely reduced Cd in the blood and tissues. What would be the possible mechanisms behind? Are there any previous studies on similar effects of EGCG or green tea's polyphenols? Are there any previous investigations indicating the detoxification of heavy metals by EGCG?
Thank you for your comment. The specifications required were added to the lines 336-346. The citations in parentheses are in the references section of the manuscript.
The paragraph added: “EGCG promotes the elimination of Cd from the blood by binding to this heavy metal through hydroxyl groups, thereby inactivating Cd through chelation. This process involves the -OH groups located on rings B and D, with the latter being the gallate group (23, 35), reacting with the free radicals generated by Cd to form a stable ring-shaped structure called a cyclic chelate. This molecule increases solubility in water, facilitating its excretion in urine and eliminating Cd from the blood and tissues (23, 36). This Cd chelation mechanism is due to the presence of eight hydroxyl groups in the EGCG structure, located in positions 3', 4', and 5', with a galate fraction in C-3. This makes it a better electron donor and, therefore, a better free radical scavenger (26, 37). These findings suggest that EGCG may improve Cd excretion from the body, offering a potential strategy to reduce Cd burden.”
- Since EGCG reduced the body's Cd concentration to the control level, all other data regarding the testis and epididymis would not make much sense. There would be no effects because of no Cd in the circulation. The authors should explore the reduction/detoxification of Cd by ECGC.
Thank you for your comment. The specifications required were added to the lines 342-346. The citations in parentheses are in the references section of the manuscript.
The paragraph added: “This Cd chelation mechanism is due to the presence of eight hydroxyl groups in the EGCG structure, located in positions 3', 4', and 5', with a galate fraction in C-3. This makes it a better electron donor and, therefore, a better free radical scavenger (26, 37). These findings suggest that EGCG may improve Cd excretion from the body, offering a potential strategy to reduce Cd burden.”
- This study aims to investigate the mitigating effects of EGCG on the toxicity of Cd. Then the Introduction should provide the background of both the toxicity of Cd and the effects of EGCG. Please provide more information on previous studies of the toxicities of Cd and the bioactivities of polyphenols. Then, clearly state the aims of this study. Otherwise, all these descriptions regarding the prepubertal development of the male reproductive system are rather pointless. Please rewrite the Introduction accordingly.
Thank you for your review and comments. This information has been added to section of 1. Introduction, in the lines 85-106. The citations in parentheses are in the references section of the manuscript.
The paragraph added: “Cd causes damage to the structure of the testicular and epididymal epithelium, such as cellular disorganization and epithelial desquamation, loss of germ cells, tubular degeneration with vacuoles, apoptosis of spermatogenic cells (17, 22), lack of adhesion between principal cells and basal cells, and a reduction in testosterone concentration (22). To counteract these effects, various antioxidant molecules have been used, such as the (-)-epigallocatechin-3-gallate (EGCG), a polyphenolic compound and catechin found in the young leaves, shoots, and stems of the Camellia sinensis plant. It makes up 50 to 80 % of the total catechin content of green tea and is the most active compound (23). EGCG has been reported to reduce the toxicity of heavy metals like nickel (Ni), arsenic (As), lead (Pb), mercury (Hg), and Cd in tissues of organs such as the testicles, liver, kidneys, and neuronal cells (23, 24). One of the properties attributed to EGCG is its antioxidant capacity, which involves eliminating ROS and chelating ionic metals (21, 25-26). Its high antioxidant activity depends on the presence of three hydroxyl groups (-OH) in its molecular structure (21). The antioxidant mechanism involves the donation of hydrogen atoms from the -OH groups to neutralize free radicals (23). Chelation occurs when these groups react with metal ions to form stable, ring-shaped structures called cyclic chelates. This structure increases the water solubility of EGCG, allowing it to be excreted in urine and eliminating heavy metals from tissues (21, 23, 24). Thus, the Cd that impacts reproductive processes could be mitigated by EGCG, which could favor reproductive health. In the present study, the effect of EGCG on antioxidant activity and the improvement of testicular and epididymal histology in cadmium-exposed prepubertal rats was evaluated.
- Taking account of the two fundamental points above, the Discussion needs to be revised.
We appreciate your suggestions, and the changes have been made throughout the manuscript. In the Discussion section, the requested information has been added.
Minor points
- Please change the group names to Control, Cd, EGCG, and Cd+EGCG in the entire text, figures, and tables.
We appreciate your suggestions, and the changes were made throughout the manuscript, as well as in the graphs and tables, so that they now read Ctrl, Cd, EGCG, and Cd+EGCG.
- Line 431, what does 'Animals that did not meet this criterion were excluded' mean? Does any group consist of fewer than 7 rats because 'not meeting this criteria'? Otherwise, please delete this sentence.
We appreciate your review and comments. The change was made in section 4.1, Experimental animals, lines 462-464. The 28 male Wistar rats included in this study had a body weight of 36.7 ± 4.2 g at 21 (PND). The animals were obtained from the vivarium facilities of the
- Figure 7, and Lines 447-451, were the chemicals injected on PND 49? If so, when (how long after) were the animals killed after the final injection?
We appreciate your review and comments. The change was made in section 4.2 in Treatments. The last administration was performed on day 49 PND, and after two hours, euthanasia was performed on all groups.

Reviewer 5 Report
Comments and Suggestions for Authors
In the present study, attempts were made to investigate the effects of cadmium (Cd), epigallocatechin-3-gallate (EGCG), or Cd+EGCG on blood testosterone levels and antioxidant enzyme activity in the testicular and epididymal structures of the prepubertal rats The manuscript is well-structured and well-planned, however, there are some comments that need to be addressed before the manuscript can be considered for publication.
Comments are as follows:
1. Rewrite the objective of the study to improve clarity (L78-80). The rats were exposed to 2 different compounds (Figure 7).
2. There should be consistency in the presentation of the Results. Present the results in tabular forms or column charts throughout the study.
3. The Reviewer suggests that the significant differences shown in the Tables and Figures are poorly presented – they lack clarity. Regardless of the presentation form, the Authors should consider to indicate the significant difference among the same groups using letters for the multiple pairwise comparison test (Tukey-Kramer), for example, for Table 1 - control, Cd, EGCG and Cd-EGCG-treated groups will be a, b, a, a, respectively (different letters within the same row are significant at P < 0.05).
4. Should consider to give the appropriate references for the euthanasia procedure, biochemical analysis, testosterone measurement, MDA measurement, etc. (L442-543). Give the procedure used for the measurement of total protein content.
5. Why did the measurements of reactive oxygen species (ROS) were not considered in this study?
6. Provide the full names for all the abbreviations/synonyms when used for the first time in the text, for examples, PND (L64) and Cd (L69).
Author Response
Response to Reviewer 5.
We thank the Editor and Reviewers for their careful reading of the manuscript in this review and their helpful suggestions for clarification and improvement.
In the present study, attempts were made to investigate the effects of cadmium (Cd), epigallocatechin-3-gallate (EGCG), or Cd+EGCG on blood testosterone levels and antioxidant enzyme activity in the testicular and epididymal structures of the prepubertal rats The manuscript is well-structured and well-planned, however, there are some comments that need to be addressed before the manuscript can be considered for publication.
Comments are as follows:
- Rewrite the objective of the study to improve clarity (L78-80). The rats were exposed to 2 different compounds (Figure 7).
We appreciate your review and comments. The change was made in Figure 7 to improve clarity in line 495.
- There should be consistency in the presentation of the Results. Present the results in tabular forms or column charts throughout the study.
We appreciate your comments. The results are presented in this manner with the understanding that they will be described in tables and figures.
- The Reviewer suggests that the significant differences shown in the Tables and Figures are poorly presented – they lack clarity. Regardless of the presentation form, the Authors should consider to indicate the significant difference among the same groups using letters for the multiple pairwise comparison test (Tukey-Kramer), for example, for Table 1 - control, Cd, EGCG and Cd-EGCG-treated groups will be a, b, a, a, respectively (different letters within the same row are significant at P < 0.05).
We appreciate your comments regarding the lack of clarity in the representation of significant differences between the same groups. We have made the requested changes by changing the interpretation of significant differences from symbols to letters, as follows: aindicates a significant difference (p < 0.05) between Ctrl vs Cd; bindicates a significant difference (p < 0.05) between Cd vs EGCG, cindicates a significant difference (p < 0.05) between Cd-exposed vs Cd+EGCG. One-way ANOVA followed by Tukey-Kramer test
- Should consider to give the appropriate references for the euthanasia procedure, biochemical analysis, testosterone measurement, MDA measurement, etc. (L442-543). Give the procedure used for the measurement of total protein content.
We appreciate your review and comments. Changes were made in section 4.4.3, Quantification of Malondialdehyde (MDA), lines 539 to 540, adding the determination of protein concentrations. The protein concentration was determined by the Bradford technique (Bradford, 1976).
- Why did the measurements of reactive oxygen species (ROS) were not considered in this study?
We appreciate your review and comments. We will take this into consideration for future research.
- Provide the full names for all the abbreviations/synonyms when used for the first time in the text, for examples, PND (L64) and Cd (L69).
We appreciate your review and comments. The requested changes to the abbreviations throughout the manuscript have been included, as well as in the Abbreviations section on line 636.

Round 2
Reviewer 2 Report
Comments and Suggestions for Authors
All inquiries were addressed by authors.
Reviewer 3 Report
Comments and Suggestions for Authors
The paper is acceptable.